# Assessing the impact of online postal self-sampling for sexually transmitted infections on health inequalities, access to care and clinical outcomes in the UK: protocol for ASSIST, a realist evaluation

Jo Gibbs [1], Alison R Howarth [1], Jessica Sheringham [2],
Louise J Jackson [3], Geoff Wong,[4] Andrew Copas [1], David J Crundwell,[5]
Catherine H Mercer,[1] Hamish Mohammed,[6] Jonathan Ross,[7] Ann K Sullivan [8],
Elizabeth Murray [9], Fiona M Burns [1]

For numbered affiliations see end of article.

**Correspondence to**
Dr Jo Gibbs; jo.gibbs@ucl.ac.uk

## ABSTRACT

**Introduction** The past decade has seen a rapid increase in the volume and proportion of testing for sexually transmitted infections that are accessed via online postal self-sampling services in the UK. ASSIST (Assessing the impact of online postal self-sampling for sexually transmitted infections on health inequalities, access to care and clinical outcomes in the UK) aims to assess the impact of these services on health inequalities, access to care, and clinical and economic outcomes, and to identify the factors that influence the implementation and sustainability of these services.

**Methods and analysis** ASSIST is a mixed-methods, realist evaluated, national study with an in-depth focus of three case study areas (Birmingham, London and Sheffield). An impact evaluation, economic evaluation and implementation evaluation will be conducted. Findings from these evaluations will be analysed together to develop programme theories that explain the outcomes. Data collection includes quantitative data (using national, clinic based and online datasets); qualitative interviews with service users, healthcare professionals and key stakeholders; contextual observations and documentary analysis. STATA 17 and NVivo will be used to conduct the quantitative and qualitative analysis, respectively.

**Ethics and dissemination** This study has been approved by South Central – Berkshire Research Ethics Committee (ref: 21/SC/0223). All quantitative data accessed and collected will be anonymous. Participants involved with qualitative interviews will be asked for informed consent, and data collected will be anonymised.

Our dissemination strategy has been developed to access and engage key audiences in a timely manner and findings will be disseminated via the study website, social media, in peer-reviewed scientific journals, at research conferences, local meetings and seminars and at a concluding dissemination and networking event for stakeholders.

## STRENGTHS AND LIMITATIONS OF THIS STUDY

⇒ Using a mixed-method, realist evaluation will provide understanding of what works for whom in which context for this complex digital health intervention.
⇒ The realist approach will enable the upheaval of the COVID-19 pandemic, and its impacts on data access, quality and implementation processes, to be included in the evaluation.
⇒ Findings from the economic evaluation will provide vital information on the cost-effectiveness and impact on health equity of online postal self-sampling services.
⇒ Findings from the implementation evaluation will inform future service delivery.

## INTRODUCTION

Sexual health remains a public health priority, with sexually transmitted infections (STIs) causing significant morbidity, substantial public health cost and contributing to antimicrobial resistance globally.[1] STIs disproportionately affect people with barriers to accessing services[2 3] and large health inequalities exist, with young people, lesbian, gay, bisexual, trans, queer or questioning (LGBTQ+), ethnic minorities and those living in more deprived areas disproportionately affected by poor sexual health.[4]

Sexual health service provision in the UK has suffered due to fragmented commissioning and service delivery,[5 6] compounded by large scale disinvestment in sexual health, at a time of increasing unmet need.[5] The concerning trends and inequalities in access to sexual healthcare prompted the UK House of Commons Health Select Committee in 2019, to recommend a new national strategy

in England to improve access to STI prevention, screening and treatment, which the Department of Health and Social Care committed to publishing in Summer 2022.[7]

The overwhelming consensus is that all healthcare systems must embrace digital technology to achieve better health, better care and reduce costs.[8] Despite substantial investment in development, the successful implementation of digital health interventions requires sociocultural change as well as technical delivery. Many such interventions lack robust evaluation against health rather than process outcomes.[9] In addition, where a digital innovation has been shown to work, there are challenges in scaling it up and building the infrastructure to support implementation across a whole healthcare system.[10]

In line with the National Health Service (NHS) digitalhealth and self-managed health strategies in England,[11] novel models of care have been introduced in response to increased demand and reduced resources.[5] These include online postal self-sampling (OPSS) for STIs and HIV, where people can order a self-sampling kit online or in a healthcare setting, take the samples themselves and then post the samples back to a laboratory. The inclusion of OPSS services within service specifications is recommended within English national guidance.[12] In recent years, there has been an active drive to divert people seeking STI test who are asymptomatic to online services, with clinic-based services reduced or centralised in many areas.[5] The expansion in provision and uptake of OPSS has accelerated because of the COVID-19 pandemic, with doubling of online consultations between 2019 and 2020 (511 979–1 062 157) and the proportion of chlamydia tests in 15–24 years from OPSS rising from 21% in 2010 to 40% in 2020.[13]

The implementation of OPSS as a commissioned service within England has yet to be evaluated. Most of the knowledge of the impact of OPSS services currently comes from observational or exploratory studies, of variable quality, evaluating a single site or OPSS service provider.[14] Much of the focus has been on assessing uptake rather than clinical (eg, treatment, partner notification) outcomes or cost-effectiveness. Assessing the impact of online postal self-sampling for sexually transmitted infections on health inequalities, access to care and clinical outcomes in the UK (ASSIST) is a widescale evaluation that aims to assess the impact of OPSS services on access to care, clinical, public health and economic outcomes, and health inequalities and to identify the factors that influence the implementation and sustainability of OPSS services. As well as informing service provision and policy within the UK, this study will also provide a critical contribution to the global research agenda for self-care interventions by answering identified key points relating to self-sampling for STIs and HIV.[15]

## METHODS AND ANALYSIS
### Study design and setting
ASSIST is a 39-month theoretically informed mixed-methods realist evaluation of the introduction of OPSS services in England (1 January 2021 to 31 March 2024).

It will provide in-depth evaluation of three case study areas (CSAs) (Birmingham, London and Sheffield) that have implemented OPSS services at different times and using different models,[16] within a national context. The CSAs have been selected to maximise diversity in geography, demographics and time since introducing OPSS services. The areas all serve diverse populations in terms of socioeconomic status and proportion of people from ethnic minorities, LGBTQ+ and young people, enabling investigation of the impact of OPSS in conjunction with the wider determinants of health. Within London, two commissioned areas have been purposively selected which have high representation of our populations of interest (as above) and a high score on the index of deprivation.

To inform the realist evaluation, this study incorporates an impact, economic and implementation evaluation, which are summarised in figure 1 and table 1 below. The impact evaluation will examine the effect of introducing OPSS on service access, clinical and public health outcomes, and health inequalities using quantitative analysis of existing surveillance data, clinic/OPSS datasets, and qualitative interviews with service users and healthcare professionals. The economic evaluation analyses the costs and outcomes associated with OPSS services compared with clinic-based services, through undertaking an economic analysis based on the resource use, clinical outcomes and cost data collected from each CSA. It will also evaluate the health equity impacts associated with OPSS in each CSA. The implementation evaluation will describe and evaluate how implementation processes and service delivery models contribute to observed variation in clinical effectiveness and cost-effectiveness.

### Overarching theoretical framework
This study uses realist evaluation, a theory driven form of evaluation which focuses on explaining how and why interventions produce outcomes under different contexts.[17] Explanations are expressed in the form of context, mechanism, outcome configurations, that explicitly link the influence of context on mechanisms which then produces outcomes.[17] The realist approach is well suited to this project because of the variation in implementation and delivery of OPSS in England at different time points. It is possible that different OPSS services, in different settings and delivered in diverse ways may produce different outcomes and realist evaluation is an ideal approach to use for such heterogeneous interventions.

Our initial programme theory (see table 2) (developed from our logic model—see figure 2) seeks to explain impact and implementation based on existing evidence[14 18] and the collective knowledge of the ASSIST research group. We will iteratively use and synthesise quantitative and qualitative data to develop and refine our initial programme theories[19] over the course of our evaluation of OPSS services, allowing us to say 'what works, for whom and under what circumstances'.

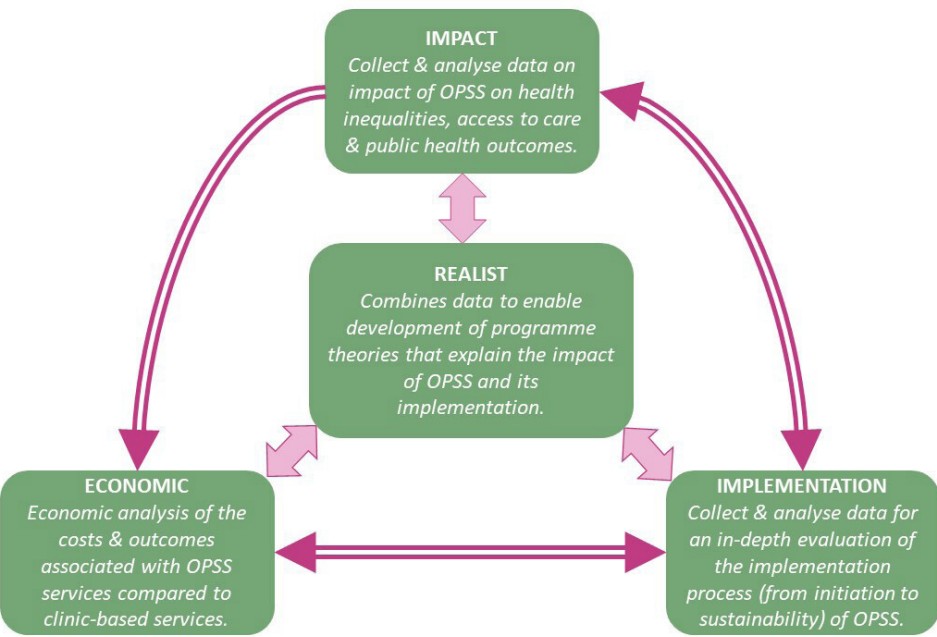

**Figure 1** ASSIST (Assessing the impact of online self-sampling for STIs & HIV) study design. OPSS, online postal self-sampling.

## Methods of data collection and sampling

The methods of data collection are summarised in table 1. We describe each type of evaluation in turn.

## Impact evaluation

### Quantitative assessment

*Data sources*

Anonymised, electronic routinely collected data from the GUMCAD STI Surveillance System (https://www.gov.uk/guidance/gumcad-sti-surveillance-system), CTAD Chlamydia Surveillance System (https://www.gov.uk/guidance/ctad-chlamydia-surveillance-system) and national HIV and syphilis self-sampling service (https://www.gov.uk/government/publications/national-hiv-self-sampling-service) datasets for the following time frames:

► January 2015–February 2020 (ie, pre-COVID).
► March 2020–December 2022.

Anonymised data on clinical outcomes will be collected from routine information held by clinic-based and online service providers in the CSAs. In each area, service user information is collected on electronic patient record systems and website databases. Clinic-based data will be collected for the following from January 2015 (or 12 months prior to implementation of OPSS if this is earlier) to December 2022. Data from online providers will be collected from the initiation of OPSS service to June 2022 and December 2022, for interim and full analyses, respectively.

Surveillance data from national datasets will be requested from the UK Health Security Agency and Office for Health Improvement and Disparities. Clinic-level and online service-level data will be requested from local data managers at each CSA.

### Primary outcomes

The primary outcomes for the impact evaluation are chlamydia, gonorrhoea, syphilis and HIV testing activity, chosen because they are epidemiologically and clinically important, captured by routinely collected data and sufficiently common to detect change over time. The analysis will determine whether the introduction of OPSS services is associated with overall changes in these activity measures and with differential change according to population characteristics (including age, gender, ethnicity, sexuality and Indices of Multiple Deprivation (IMD)), reflecting a change in health inequality.

### Power analysis

Three contrasting settings have been selected to serve as case studies. Our key objective which is most 'demanding' of the data, and hence drives the power calculation, is the detection of differences in the change in the primary outcomes after introduction of OPSS by key population characteristics. Data will be obtained over a continuous period before, during and after implementation of OPSS. However, to simplify the power calculation (and to be conservative) the comparison of 1-year periods before and after OPSS are considered (different for each area).

As an example, analysis of data from one area is considered—Birmingham, and the testing rate for STIs excluding chlamydia for which data are publicly available. The rate changed from 193 per 1000 population aged 15–64 years in 2014 (pre OPSS) to 198 per 1000 in 2016 (post OPSS). With around 140 000 tests per year there is more than 99% power to detect a change as small as 1% in the proportion of those testing with a particular

**Table 1**  Summary of primary objectives, data collection methods and reason for using data collection methods

| Evaluation | Primary objectives | Data collection method | Rationale of data collection method |
|---|---|---|---|
| Impact | Establish what has been the change in access to care and service delivery as a consequence of the introduction of OPSS.<br>Determine the impact of OPSS services on health inequalities and key clinical and public health outcomes.<br>Determine who is accessing online services and clinic-based services, in what context, and why.<br>Explore user and provider experience of OPSS services. | 1.Quantitative:<br>1.1 Routinely collected national data (eg, UKHSA data)<br>1.2 Clinic and OPSS service datasets<br>1.3 Large population surveys (eg, Natsal)<br>2.Qualitative:<br>2.1 Interviews with previous users of OPSS (10–15/CSA), users of clinic-based (10–15/CSA), and users of both OPSS & clinic-based services (10–15/CSA)<br>2.2 Healthcare professional interviews | To determine the demographic and key clinical outcomes of people accessing OPSS and clinic-based service, and how this has changed over time.<br>Analyses of detailed behavioural and biological data within each CSA preimplementation and postimplementation of OPSS services to evaluate triage/safe-guarding systems and impact on wider sexual health needs. 1.3 will help contextualise health inequality and clinical findings. |
| Economic | Analyse the costs and outcomes associated with OPSS services compared with clinic-based services<br>Explore impacts on health equity associated with different models of service provision | Resource use and cost data from clinic and OPSS service datasets Information on unit costs or prices will be sourced to attach to each resource use item using published information.[30] Where necessary, local cost information for each area will also be obtained from accounting systems within finance departments, service and finance leads, and laboratory managers.<br>Data on patient resource use and clinical outcomes will be collected as per the impact evaluation. | Resource use and cost data will be collected from each area to estimate the overall costs associated with OPSS services compared with clinic-based services. |
| Implementation | Identify, characterise and understand the following implementation factors and how they relate to observed variation in uptake, use, clinical outcomes, costs and overall impact on health inequalities and public health:<br>► Key contextual factors for each case study area<br>► Planned and actual implementation interventions in each case study area<br>► Stakeholder perceptions of key factors influencing service delivery, acceptability and observed outcomes | 3.1 Contextual drivers<br>► Document analysis<br>► Key informant/stakeholder interviews (5–10/CSA)<br>3.2 Planned and actual implementation processes<br>► Semi-structured interviews with commissioners (six across all areas), the tendering team (8–10 across all areas), clinical leads, service managers and healthcare professionals<br>► Contextual observation (3–5/CSA) | The implementation evaluation is divided into two sections which seek to address its objectives to identify and understand<br>1. key contextual factors for each case study area;<br>2. planned and actual implementation interventions in each case study area, and stakeholder perceptions of key factors influencing service delivery, acceptability and observed outcomes. |
| Realist | Bring together the data from each workstream into a coherent whole; it will ensure that the initial programme theories of impact and implementation of OPSS are iteratively refined into more detailed realist programme theories using relevant data from across all workstreams | Data to inform our interpretation of the relationships between contexts (C), mechanisms (M) and outcomes (O) will be sought across the different data sources from each workstream (eg, mechanisms inferred from one source could help explain the way contexts influenced outcomes in a different source). | Synthesising data from different sources is often necessary to compile CMO configurations, since not all parts of the configurations will always be found in the same source. |

CSA, case study area; Natsal, National survey of sexual attitudes and lifestyles; OPSS, online postal self-sampling; UK HSA, UK Health Security Agency.

characteristic (eg, ethnic minority group or gender) whatever the proportion before OPSS.

### Data analysis

For each CSA and primary outcome separately:

► Change will be analysed after introduction of OPSS in the overall number of tests and the proportion of those testing with particular population characteristics.
► Change will be expressed in the rate of testing per year and, using estimates of the catchment area population size, change in the rate of testing per 1000 population per month, using Poisson regression.
► Change will be formally assessed in health inequality by testing for differential change over time (pre and post OPSS) by population characteristic (eg, gender, ethnic minority group) through including interaction terms in the regression models.
► Data will be pooled across the three areas to test for different change between areas.

**Table 2** Initial programme theory: set out as assumptions of what is expected of the OPSS intervention

| Service users | Commissioners and providers |
|---|---|
| ► Easy to find and access<br>► Convenient<br>► Easy to use for (fits in with 21st Century life)<br>► Provides privacy and minimises embarrassment/judgement by others<br>► As 'good' as face to face<br>► Self-sampling is easy to do, and people are willing to do these<br>► People believe the results of the self-sample | ► Saves money<br>► Frees up capacity (efficient way to manage asymptomatic service users)<br>► Saves time for providers<br>► Provides the additional capacity to deal with demand<br>► Everyone has equal access |

OPSS, online postal self-sampling.

Anonymised data from Natsal-Covid study[20–22] and Natsal-4[23] will be accessed. Natsal, the National Surveys of Sexual Attitudes and Lifestyles, are nationwide surveys of sexual behaviour designed to be broadly representative of the general population that have been run approximately every 10 years since 1990. Analyses of Natsal-3, Natsal-Covid and Natsal-4 will provide insight of change over time in access to sexual health services, testing, diagnoses and sexual behaviour within the general population, and will provide context as to whether change is greater in areas that fully implemented OPSS services than others.

To address the possibility of confounding arising from other service changes that may have occurred at a similar time to OPSS, and the immediate impact of the COVID-19 pandemic, an analysis will be conducted based on an in-depth understanding of the health service provision over the full range of time 2014–2022 in our CSAs. By collecting the dates of other changes in provision and prespecifying the likely lag for these changes to affect our outcomes, Poisson regression models can be developed for our outcomes in continuous time that permit an interrupted time series analysis to attempt to address the

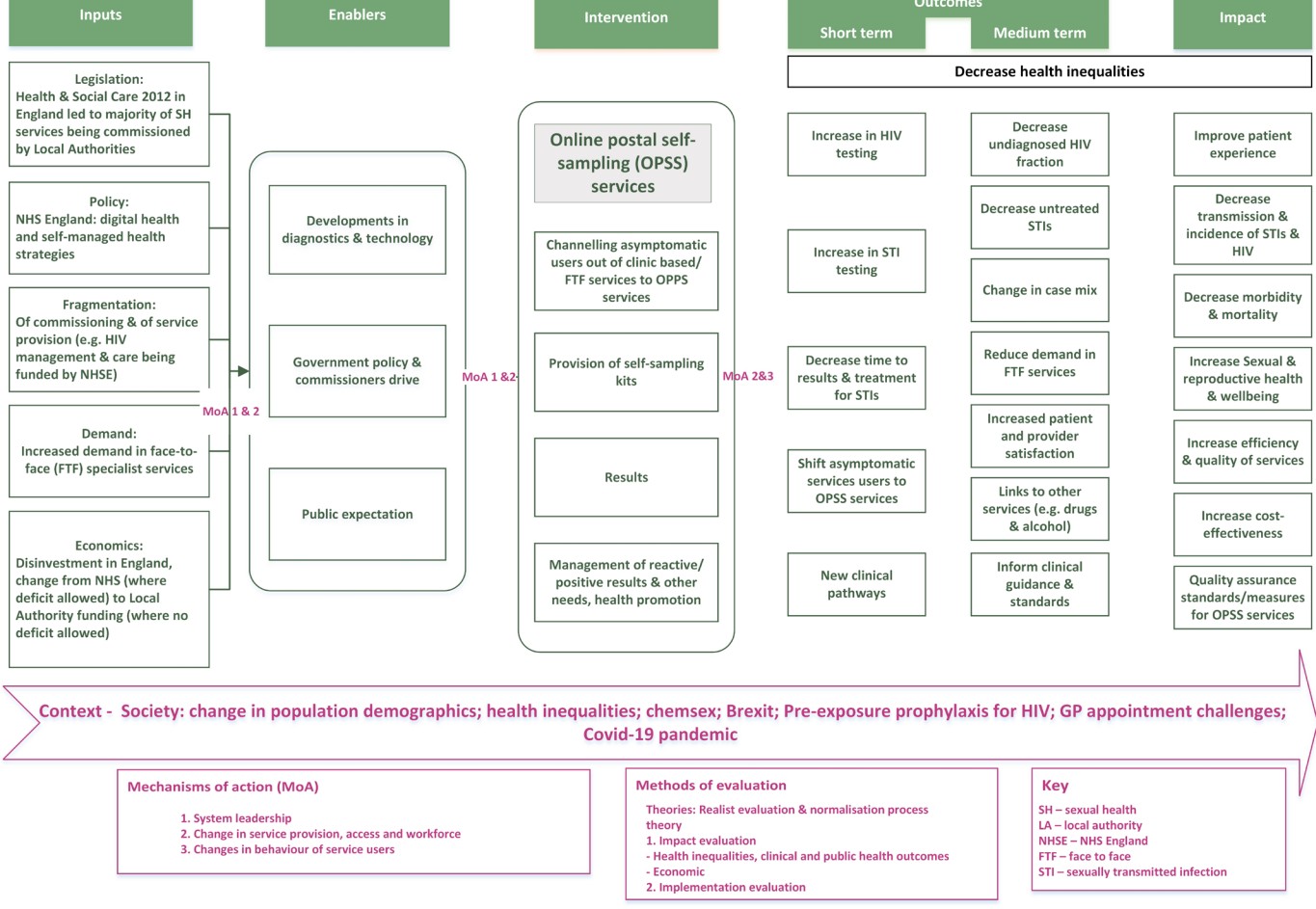

**Figure 2** Logic model explaining the introduction of OPSS services.

specific effect of OPSS adjusting for the effects of other changes. Our primary interest is in assessing the impact of including OPSS alongside clinic testing in 'normal conditions' rather than the impact of the rapidly scaled up use of OPSS that occurred during the first year of the COVID-19 pandemic, which has since decreased but remained higher that in pre-COVID times.[13]

### Acceptability of OPSS and clinic-based services

A mixed-methods approach (summarised in table 1) will be taken to explore and understand the impact of the introduction of OPSS services on acceptability of sexual health services (OPSS and clinic-based), user and provider experience and user requirements of sexual health services, as well as why some potential OPSS users opt to use clinic-based services.

#### *Quantitative analysis*

The uptake of each component of the intervention will be analysed using logistic regression modelling for each component, stratifying by age, gender, sexuality, ethnicity and IMD.

#### *Qualitative recruitment*

Clinic-based service users: potential participants will be identified and recruited by a member of the healthcare team on the day of their consultation. Targeted responsive recruitment of specific patient profiles within clinics will include understanding and tailoring recruitment processes to local multidisciplinary teams, providing regular feedback on relative recruitment success, and iteratively developing responsive action plans.

Online service users: potential participants will be recruited via (1) information posted on the OPSS website with a link taking them to a secure online form where they will be asked to complete their contact details and screening information; and (2) advertisements using social media and the study website, and local community settings. The online information will explain that not all interview volunteers will be contacted.

Recruitment numbers are summarised in table 1. Up to 135 service users will be recruited across the three CSAs. A member of the research team will contact potential participants by telephone, email or encrypted online messaging service (eg, WhatsApp) (according to preference). The researcher will determine eligibility, explain the study and arrange a convenient time for the interview. Tailored participant information sheets will be provided to service users by email, post or a via a link to the study website. Service users will be reimbursed £30/interview.

Healthcare providers: potential participants will be invited by email to take part. Up to 45 healthcare providers will be recruited across the three CSAs.

#### *Qualitative data collection*

A topic guide was developed, reviewed by public and patient involvement representatives and piloted before use with study participants, with versions created specific to service users and healthcare professionals (see online supplemental files 1–3). Iterations to the topic guides will be made, informed by findings emerging from early interviews.

#### *Qualitative analysis*

All qualitative data will be transcribed verbatim by professional services, fully anonymised and entered into NVivo for coding.

The analysis will be multistaged. Preliminary review of the interviews, through the use of a rapid reflection tool adapted from 'RREAL' RAP sheets,[24] will inform iterative refinement of both the programme theory and topic guides. Stage 1 of the full analysis will use inductive thematic analysis to identify emergent themes around the characteristics of the users, provider, decision making and the OPSS journey. Data will also be analysed deductively into themes informed by constructs from mid-range theories. The choice of theory will depend in part on the fit with inductive codes, but might include: Levesque's theory of access at the interface of populations and health systems,[25] and the theory of candidacy, by Dixon-Woods *et al*,[26] which was developed to understand access to healthcare by vulnerable groups. To explore the correspondence to workstream 3 (Implementation), we will also explore the relevance of middle-range theory constructs applied in this workstream.

Stage 2 will use a realist logic analysis to build contexts (C), mechanisms (M) and outcomes (O) configurations (CMOCs) from the data within and across the themes generated. In this stage, data will also be sought from within and across the themes to inform our interpretations of where the CMOCs developed fit within our initial programme theory—thus gradually refining it.

Where possible and relevant, observed variation in quantitative outcomes collected for the impact evaluation will also be used to develop and refine CMOCs.

### Economic evaluation

An economic evaluation will be undertaken to compare the costs and benefits for screening asymptomatic individuals undertaken using self-sampling kits ordered online, compared with screening of the same group in a clinic setting, across the CSAs. As a secondary analysis, the potential impacts on health equity associated with different types of service provision will be assessed.[27] The economic evaluation will be conducted and reported in accordance with relevant guidelines.[28 29]

#### *Cost–consequence analysis*

Initially, a cost–consequence analysis will be presented, which involves reporting all costs and outcomes in a disaggregated manner.[29] Resource use and cost data will be used to estimate the overall costs associated with OPSS services compared with clinic-based services. Information on unit costs or prices will be sourced to attach to each resource use item using published information.[30] Where necessary, local cost information for each area will also be obtained from accounting systems within finance

departments, service and finance leads, and laboratory managers. An incremental economic analysis will be conducted using the primary outcome of cost per positive case identified and the secondary outcomes of cost per patient screened, cost per patient treated and cost per partner identified/treated (if data quality on partner notification permit this). The economic component will explore how different service configurations can be used to achieve the optimal level of health benefit, within existing resource constraints. This element will identify and model patient pathways across the CSAs, assess comparative costs and outcomes, and analyse different scenarios for service configuration.

### Analysis of impacts on equity

Currently these methods are being refined and a range of possible methods will need to be considered.[27] For example, recommended approaches include equity impact analysis (analyses distributional impacts on different groups) and equity trade-off analysis (examining trade-offs between improving total health and reducing health inequality).[31] Variations of multicriteria decision analysis have also been proposed as possible methods.[32] As such methods have not previously been used in a sexual health context, a review of the literature will be conducted to assess the most appropriate approach and be informed by emerging practice.[33 34] This approach will then be applied as a secondary analysis to allow decision-makers to access both a traditional analysis and a fuller analysis taking into account equity considerations.

Deterministic and probabilistic sensitivity analyses will be conducted to explore the effects of the uncertainty in the parameter estimates on the results.[35] Deterministic sensitivity analysis involves varying one or more parameters while keeping the others at their baseline value. A probabilistic sensitive analysis involves varying all parameters simultaneously, and multiple sets of parameter values are sampled from defined probability distributions.[36]

### Implementation evaluation

Our candidate theory for the implementation evaluation is normalisation process theory (NPT),[37] a substantive middle-range theory that focuses on the work required for initiating, integrating and embedding (normalising) OPSS into routine practices.[9 38] NPT recognises that normalisation is a non-linear, iterative and contingent process, and influenced by the contexts in which implementation occurs.

Document analysis, interviews with key informants/ stakeholders (including commissioners, clinicians, health advisors, voluntary sector) and contextual observation will identify and map the intervention components and actions required for incorporation and normalisation within a service. The factors that influence this and OPSS outcomes will be identified, for example, characteristics of the local and national contexts and changes in patient flow (see logic model, figure 2).

The document analysis will use resources such as (1) national and local policy and local authority minutes; (2) service specification, strategy documents and consultation documents; and (3) service-level minutes from management meetings and consultant meetings that describe the decisions around how OPSS was provided and sustained.

Key informant/stakeholder interviews will explore the decision to offer OPSS as part of a service (historical), in addition to contemporaneous perspectives on the contextual factors perceived to influence the implementation process. They will examine the drivers for starting and continuing to provide OPSS, and factors that enabled or inhibited the set-up and continuation of the service, using a topic guide informed by NPT.[37] Up to 58 semistructured interviews will be conducted with commissioners, the tendering team, clinical leads, service managers and clinical healthcare professionals who came together to decide how to deliver OPSS. Interviews will explore the provider experience of implementing, working within and with OPSS services, including the processes devised and actually implemented, and their impact on initiation embedding and integration.

Contextual observations will involve brief periods of work shadowing with a small number of healthcare professionals and administrative staff to better understand the actions and adaptation required to fit OPSS with work practices and their lived experiences of providing the OPSS journey. Up to 15 contextual observations will be conducted across the three CSAs. They will comprise:

1. In-person observation of healthcare professionals during clinical consultations.
2. Think aloud exercises about OPSS consultation scenarios with healthcare professionals.
3. Observation and think aloud exercises with administrative staff about managing clinical records and/or administrative tasks.

### Key informant/stakeholder recruitment

Participants will be invited by email to take part in key informant interviews and contextual interviews.

### *Data analysis*

Initial coding will be largely inductive, though informed by initial programme theory assumptions and the study research questions.[39] Then, guided by others' experience of applying normalisation process frameworks (eg, Macfarlane *et al*[40]), we will investigate alignment with NPT codes.[39] This stage will seek to draw out prominent themes corresponding to our understanding of higher order constructs of NPT—context, coherence, cognitive participation, collective action, reflective monitoring and implementation outcomes—as they apply to ASSIST.

### Development of programme theories

Data from all three components will also be analysed using a realist logic of analysis aimed at developing iterative refinements of the initial programme theory to include CMOCs. Where possible and relevant, observed variation

in quantitative outcomes as collected in the impact evaluation (eg, in uptake, use, clinical and economic outcomes, and impact on health inequalities) will be used to develop and refine our context mechanism and outcome configurations. The realist analysis to support the formation of CMO configurations and the analyses in each evaluative component will be iterative, rather than conducted sequentially.

Interpretive cross-case comparison will be used to understand and explain how and why observed outcomes have occurred, for example, by comparing how outcomes may differ according to population group or service model, to understand how context, problem or diversity have influenced findings.[41] Where appropriate, the following forms of reasoning will be used to make sense of the data:

► Juxtaposition: for example, where data about uptake of OPSS in one source enables insights into data about uptake in another source.
► Reconciliation: where data differ in apparently similar circumstances, further investigation is appropriate in order to find explanations for these differences.
► Adjudication: on the basis of whether threats to the validity of data in one source might make us question their trustworthiness compared with data from another source.
► Consolidation: where outcomes differ in particular contexts, an explanation can be constructed of how and why these outcomes occur differently.

The evaluation will move iteratively between the analysis of particular examples, refinement of programme theory, and further data collection to test particular parts of our programme theories. This will allow us to say 'what works, for whom and under what circumstances' for the impacts and in implementing and sustaining OPSS services.

## PATIENT AND PUBLIC INVOLVEMENT AND ENGAGEMENT

ASSIST has engaged with meaningful public and patient involvement and engagement (PPIE), including a lay member (DC), coapplicant and PPIE lead, who has been actively involved from the start of the study. The British Association for Sexual Health & HIV (BASHH) and the Terrence Higgins Trust (one of the UK's leading HIV and sexual health charities) have established a joint Lay Research Panel, comprising a diverse range of lay reviewers who have received training in peer review. The panel reviewed the ASSIST lay summary and was strongly supportive of this study. Links have also been developed with NAZ (a charity focusing on sexual health improvement and HIV support services for ethnic minority communities), as the impact on ethnic minority communities will be one of our key outcomes. The BASHH/THT Lay Panel and NAZ welcomed our focus on young people, LGBQT+, and ethnic minority groups disproportionately affected by STIs and HIV, as both organisations are concerned that online services may only attract certain communities. Our three CSAs all serve large diverse populations enabling our evaluation to determine the impact on health inequalities. The panel also had concerns around data security which will be explored within the in-depth interviews.

Individuals from the National Institute for Health and Care Research (NIHR) Applied Research Collaboration North Thames Research Advisory Panel with experience of PPIE have been recruited to sit on our Expert Advisory Group and Study Steering Committee. Recognising the challenges of recruiting PPIE for studies on sexual health, and to ensure that we do not overburden those organisations and individuals who do engage, we have been collaborating with colleagues within the Institute for Global Health at University College London to put together a cross-project community advisory panel. This group aims to recruit a representative population that will provide an ongoing PPIE resource for several studies.

As well as providing invaluable advice on the design and conduct of the study, involvement from members of the public and community groups will assist with access to people in groups particularly relevant to our study and is essential for wider engagement. PPIE will inform the tone, pitch and content of communications and the study website. Advice will be sought from the NIHR PPI Centre on our overall approach. All PPIE will be supported according to INVOLVE guidance.

## ETHICS AND DISSEMINATION
### Ethical aspects

Ethical approval for this study has been granted by South Central—Berkshire Research Ethics Committee (ref: 21/SC/0223).

Participants will be fully informed about what taking part involves before they provide consent to participate. Before the interview, the participant information sheet and consent form will be sent by to all service users who express an interest in taking part. This will inform them about the interview topic and purpose of the research, and gives them ample time to consider whether they would like to take part. Our experienced research staff are trained to assess participants' understanding of the study procedures and ability to consent. The subject matter may be delicate and possibly emotive. The interviewers have extensive experience of collecting sensitive data and are trained in strategies for dealing with participants who are uncomfortable with the interview questions. In addition, members of the research team collecting data will have a resource pack containing helpline numbers and health promotion materials in order to provide information to participants, if required.

Persons aged 16 and 17 (and above) are included in this study because the prevalence of bacterial STIs is high among young people, and people who are 16 and above are eligible to access OPSS, so it is important to understand how OPSS affects their care.

The members of the clinical care team in the participating sexual health services will determine if people are

suitable to participate in the study. Participants who indicate their willingness to be part of the research process online will not be contacted if safeguarding concerns have been flagged.

## Data protection

This study has been registered with the UCL Data Protection Office (Ref: Z6364106/2021/04/36 health research), as data will be stored in the UCL Data Safe Haven. We will comply with the requirements of General Data Protection Regulation (2016/679) and the UK Data Protection Act (2018) with regard to the collection, storage, processing and disclosure of personal information, and will uphold the Act's core principles.

## Dissemination

The six key audiences for this research are: (1) current and future service users, and members of the public; (2) service providers; (3) commissioners; (4) professional associations (eg, BASHH; (5) external statutory organisations (eg, the Care Quality Commission, NHS Digital) and (6) academia. Our dissemination strategy has been developed to access and engage all of these audiences with our findings and recommendations in a timely manner. The strategy will follow The Health Foundation's 'Communicating your research—a toolkit'[42] and leverages existing resources within the participating organisations, such as their academic infrastructure, professional relationships and community networks fully. Study findings will be disseminated via the study website and Twitter feed, a dissemination and networking event for stakeholders at the end of the project, in scientific journals, at research conferences, and at local meetings and seminars.

**Author affiliations**
[1]Institute for Global Health, University College London, London, UK
[2]Institute of Epidemiology and Health Care, University College London, London, UK
[3]Institute of Applied Health Research, University of Birmingham, Birmingham, UK
[4]Nuffield Department of Primary Care Health Sciences, Oxford University, Oxford, UK
[5]Lay representative, London, UK
[6]Blood Safety, Hepatitis, STIs and HIV Division, UK Health Security Agency, London, UK
[7]University Hospitals Birmingham NHS Foundation Trust, Birmingham, UK
[8]Directorate of HIV and Sexual Health, Chelsea and Westminster Hospital NHS Foundation Trust, London, UK
[9]Primary Care and Population Health, University College London, London, UK

**Acknowledgements** In memory of Dr Naomi Fisher, a coapplicant on the original ASSIST submission to the NIHR. Naomi helped design and write the grant proposal for ASSIST, leading on the implementation evaluation.

**Contributors** JG and FMB are co-chief investigators of the study. JG, FMB, JS, LJJ, GW, AC, DC, CHM, HM, JR and EM conceived the study and secured funding. JG, FMB, ARH, JS, LJJ, GW, AC, DC, CHM, HM, JR, AS and EM are responsible for the planning and delivery of the study. JG leads on the impact evaluation; LJJ leads on the health economics evaluation; JS leads on the implementation evaluation; and GW leads on the development of programme theories. AC is the study statistician. All authors contributed to the development of the study design and establishment of procedures. JG led on preparing the manuscript. All authors critically reviewed and approved the final version.

**Funding** This study is funded by the NIHR Health Services and Delivery Research Programme (NIHR129157). This study is sponsored by University College London.

**Disclaimer** The views expressed are those of the authors and not necessarily those of the NIHR or the Department of Health and Social Care. This report is independent research supported by the NIHR ARC North Thames.

**Competing interests** None declared.

**Patient and public involvement** Patients and/or the public were involved in the design, or conduct, or reporting, or dissemination plans of this research. Refer to the Methods section for further details.

**Patient consent for publication** Not applicable.

**Provenance and peer review** Not commissioned; externally peer reviewed.

**ORCID iDs**
Jo Gibbs http://orcid.org/0000-0001-5696-0260
Alison R Howarth http://orcid.org/0000-0002-0597-6614
Jessica Sheringham http://orcid.org/0000-0003-3468-129X
Louise J Jackson http://orcid.org/0000-0001-8492-0020
Andrew Copas http://orcid.org/0000-0001-8968-5963
Ann K Sullivan http://orcid.org/0000-0001-7532-9562
Elizabeth Murray http://orcid.org/0000-0002-8932-3695
Fiona M Burns http://orcid.org/0000-0002-9105-2441

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
