## [Reviewer comments · BMJ Open]

ARTICLE DETAILS

TITLE (PROVISIONAL)	Assessing the impact of online postal self-sampling for sexually transmitted infections on health inequalities, access to care and clinical outcomes in the UK: protocol for ASSIST, a realist evaluation
AUTHORS	Gibbs, Jo; Howarth, Alison; Sheringham, Jessica; Jackson, Louise; Wong, Geoff; Copas, Andrew; Crundwell, David; Mercer, Catherine; Mohammed, Hamish; Ross, Jonathan; Sullivan, Ann; Murray, Elizabeth; Burns, Fiona M.

VERSION 1 – REVIEW

REVIEWER	Daniel McCartney
REVIEW RETURNED	LSHTM

GENERAL COMMENTS	I appreciated the opportunity to review this interesting study protocol for ASSIST, which is comprehensive and well-written. The study represents an important and relatively under-investigated area in sexual health, which I expect will produce useful results and recommendations not only for implementation and service delivery in England (and the rest of the United Kingdom), but also for many other countries considering the introduction or currently implementing such self-sampling, STI screening models. Importantly, this study has the opportunity to contribute to the global research agenda for self-care interventions by answering some collaboratively identified research questions related to the self-collection of samples for STI testing. If not done already, I recommend reviewing Section 5 (pg. 108-109) of the "WHO guideline on self-care interventions for health and well-being, 2022 revision." A relevant example is the need to better understand the proportion of people who seek appropriate care and treatment services after receiving a positive result after the self-collection of samples for STI testing. I also wanted to share some additional comments for consideration: 1. Symptomatic vs. asymptomatic: While mentioned, I would suggest explaining more clearly the intention of OPSS for screening asymptomatic users. If not already included as a question for the user interviews, it will be interesting to know if any users were indeed symptomatic but chose to use OPSS, and if so, the reasons for this.2. LGBTQ+: This is an important aspect along with the other diverse populations and characteristics noted in the protocol. While likely intended, I would suggest being more explicit on the intended characteristics within this umbrella term (including sexual orientation, gender identity/expression, sex characteristics), and the possible
--

	overlapping identities to better understand the impact within LGBTQ+ communities. I'm confident this expertise on issues related to sexual and gender diversity exists, if not within the list of authors, within the described Lay Research Panel and other PPIE components, and I would encourage to leverage this further during the study. 3. One minor edit for clarity, in case raised by others: (Page 5, line 11) I noted an incomplete sentence: ("...implemented OPSS services at different times and using different [missing],[15] within a national..."); should this be "models"? I very much look forward to seeing the results from this study.
--	--

REVIEWER	Irith De Baetselier Institute of Tropical Medicine, Department of Clinical Sciences
REVIEW RETURNED	06-Oct-2022

GENERAL COMMENTS	Online postal self-sampling is becoming a very important pillar in the testing strategy for STIs. Indeed, in the UK, providers of such programs are increasingly available and according to a very recent systematic review, quality is often sub-optimal. I'm looking forward to the results of this important research. I made a few remarks. Abstract:  - It is unclear what is meant by the three case study areas? (found in Method section: Birmingham, London & Sheffield, I would add this to the abstract if possible) Strengths and limitations:  - I would remove complex from the first sentence. What is complex? Methods:  - The different time periods mentioned in the protocol can cause some confusion. The study itself will run from 01/01/2021 to 31/03/2024 but the data that is going to be reviewed was from 01/2015 to 04/2023 for clinic-based info and 06/22 to 04/23 for OPSS services. Maybe this could be more clearly described? Objectives/Collection methods: To assess the impact of OPSS services on health inequalities, you will use different quantitative datasets. However, are OPSS service datasets questioning income/sexual orientation etc? Do you have access to these questionnaires? Final remark: it would be interesting to end the manuscript with a discussion section where you briefly mention what has been done until today.
---

VERSION 1 – AUTHOR RESPONSE

Reviewer: 1

Dr. Daniel McCartney, LSHTM

Comments to the Author:

I appreciated the opportunity to review this interesting study protocol for ASSIST, which is comprehensive and well-written. The study represents an important and relatively under-investigated area in sexual health, which I expect will produce useful results and recommendations not only for implementation and service delivery in England (and the rest of the United Kingdom), but also for many other countries considering the introduction or currently implementing such self-sampling, STI screening models.

Thank you.

Importantly, this study has the opportunity to contribute to the global research agenda for self-care interventions by answering some collaboratively identified research questions related to the self-collection of samples for STI testing. If not done already, I recommend reviewing Section 5 (pg. 108-109) of the "WHO guideline on self-care interventions for health and well-being, 2022 revision." A relevant example is the need to better understand the proportion of people who seek appropriate care and treatment services after receiving a positive result after the self-collection of samples for STI testing.

Thank you. We have added the following sentence in the introduction:

As well as informing service provision and policy within the UK, this study will also provide a critical contribution to the global research agenda for self-care interventions by answering identified key points relating to self-sampling for STIs and HIV.

I also wanted to share some additional comments for consideration:

1. Symptomatic vs. asymptomatic: While mentioned, I would suggest explaining more clearly the intention of OPSS for screening asymptomatic users. If not already included as a question for the user interviews, it will be interesting to know if any users were indeed symptomatic but chose to use OPSS, and if so, the reasons for this.

Thank you. We agree that this is a key point. The case study areas chosen have taken different approaches in terms of access to OPSS for people with symptoms. For example, the Umbrella service, although recommending that people with symptoms access their clinic-based services, allow people with symptoms to access OPSS. Sexual Health London was developed to channel shift asymptomatic patients from clinic-based services to online testing. However, during the COVID-19 pandemic, they changed their approach to allow minimally symptomatic patients to access OPSS. Given the importance of this from an access perspective, and an individual decision-making perspective, we do explore this within user interviews.

2. LGBTQ+: This is an important aspect along with the other diverse populations and characteristics noted in the protocol. While likely intended, I would suggest being more explicit on the intended characteristics within this umbrella term (including sexual orientation, gender identity/expression, sex characteristics), and the possible overlapping identities to better understand the impact within LGBTQ+ communities. I'm confident this expertise on issues related to sexual and gender diversity exists, if not within the list of authors, within the described Lay Research Panel and other PPIE components, and I would encourage to leverage this further during the study.

Thank you. We are taking an intersectional approach, both to individuals who identify as LGBTQ+ and also to other key populations who may be disproportionately affected by the impact of OPSS services.

The authors do have expertise relating to sexual and gender identity, and we also recognise the need for lived experience of this beyond the authors and have insured this within the Lay Research Panel and PPIE components.

3. One minor edit for clarity, in case raised by others: (Page 5, line 11) I noted an incomplete sentence: ("...implemented OPSS services at different times and using different [missing],[15] within a national..."); should this be "models"?

Thank you for identifying this omission. We have added the word "models".

I very much look forward to seeing the results from this study.

Thank you.

Reviewer: 2

Dr. Irith De Baetselier, Institute of Tropical Medicine Comments to the Author:

Online postal self-sampling is becoming a very important pillar in the testing strategy for STIs. Indeed, in the UK, providers of such programs are increasingly available and according to a very recent systematic review, quality is often sub-optimal. I'm looking forward to the results of this important research. I made a few remarks.

Abstract:

- It is unclear what is meant by the three case study areas? (found in Method section: Birmingham, London & Sheffield, I would add this to the abstract if possible)

Thank you for highlighting. We have now added "Birmingham, London & Sheffield" to the Methods section of the abstract.

Strengths and limitations:

- I would remove complex from the first sentence. What is complex?

A 'complex intervention', as defined by the UK Medical Research Council, is one with several interacting components. Additionally, interventions can be thought of as complex if they are dependent on the behaviours of those delivering and receiving the intervention, there are a range of possible outcomes, or there is a need to tailor the intervention to different contexts and settings. (BMJ 2008; 337)

The introduction of OPSS meets the criteria for 'complex' on both aspects however we have amended the text to read 'complex digital health intervention'

Methods:

- The different time periods mentioned in the protocol can cause some confusion. The study itself will run from 01/01/2021 to 31/03/2024 but the data that is going to be reviewed was from 01/2015 to 04/2023 for clinic-based info and 06/22 to 04/23 for OPSS services. Maybe this could be more clearly described?

Thank you. We explain the following within the text:

- Study will run from 01/01/2021 to 31/03/2024
- Routinely collected data from national datasets will be reviewed from January 2015- December 2022
- Clinic based data will be collected from January 2015 (or 12 months prior to implementation of OPSS if this is earlier) to December 2022
- Data from online providers will be collected **from the initiation of the OPSS service to December 2022**

We have revised the final data collection point for clinic based data and data from online providers to December 2022.

Objectives/Collection methods: To assess the impact of OPSS services on health inequalities, you will use different quantitative datasets. However, are OPSS service datasets questioning income/sexual orientation etc? Do you have access to these questionnaires?

The OPSS service datasets capture sociodemographic data including age, ethnicity, gender, sexual orientation (or sexual behaviour), & postcode. Data on income is not collected (nor education level) however we will use the postcode to generate the IMD as a surrogate measure. We do have access to the questionnaires.

Final remark: it would be interesting to end the manuscript with a discussion section where you briefly mention what has been done until today.

Unfortunately, word count limitation precludes an update on progress to date or interim findings within this protocol paper.